# Circulating Serum Cell-Free Mitochondrial DNA in Amyotrophic Lateral Sclerosis

**DOI:** 10.3390/cells14181433

**Published:** 2025-09-12

**Authors:** Giada Zanini, Ilaria Martinelli, Giorgia Sinigaglia, Elisabetta Zucchi, Federico Banchelli, Cecilia Simonini, Giulia Gianferrari, Andrea Ghezzi, Jessica Mandrioli, Marcello Pinti

**Affiliations:** 1Department of Life Sciences, University of Modena and Reggio Emilia, 41125 Modena, Italy; giada.zanini@unimore.it (G.Z.); giorgia.sinigaglia@unimore.it (G.S.); marcello.pinti@unimore.it (M.P.); 2Department of Neurosciences, Ospedale Civile di Baggiovara, Azienda Ospedaliero-Universitaria di Modena, 41126 Modena, Italy; federico.banchelli@gmail.com (F.B.); ceciliasimonini24@gmail.com (C.S.); giulia.gianferrari@unimore.it (G.G.); andrea.ghezzi@unimore.it (A.G.); jessica.mandrioli@unimore.it (J.M.); 3Neuroscience PhD Program, Department of Biomedical, Metabolic, and Neural Sciences, University of Modena and Reggio Emilia, 41125 Modena, Italy; 4Department of Biomedical, Metabolic and Neural Sciences, University of Modena and Reggio Emilia, 41125 Modena, Italy

**Keywords:** amyotrophic lateral sclerosis, circulating cell-free mitochondrial DNA, biomarker, neuroinflammation, oxidative stress, mitochondria

## Abstract

Mitochondrial dysfunction is a key pathological hallmark in amyotrophic lateral sclerosis (ALS), yet the role of circulating cell-free mitochondrial DNA (Cf-mtDNA) as a biomarker remains unclear. This study aimed to investigate serum Cf-mtDNA levels in ALS patients compared to healthy controls and explore its associations with disease biomarkers, clinical progression, and survival. We conducted a case–control study measuring Cf-mtDNA levels in serum samples from 54 ALS patients and 36 age- and sex-matched healthy controls using quantitative droplet digital PCR. Correlations between Cf-mtDNA levels and clinical features, neurofilament concentrations, inflammatory indices, and survival were assessed. The average Cf-mtDNA level in ALS patients was 2,426,315 copies/mL of serum (IQR: 865000–2475000), compared to 1,885,667 copies/mL of serum (IQR: 394250–2492500) in controls (*p* = 0.308). ROC analysis yielded an AUC of 0.595 (95% CI: 0.468–0.721), indicating very limited discriminant ability. Cf-mtDNA levels were inversely correlated with serum creatinine concentrations (r = –0.335, *p* = 0.018), but showed no significant associations with ALS phenotype, disease staging, neurofilaments, inflammatory indices, or survival. These findings suggest that, in a predominantly sporadic ALS cohort, serum Cf-mtDNA may not serve as a standalone diagnostic or prognostic biomarker, in contrast to previous reports. Methodological differences, cohort composition, and genetic heterogeneity may account for these discrepancies. Our results underscore the importance of further large-scale, longitudinal studies incorporating genetic stratification and multi-biomarker approaches to better elucidate the role of Cf-mtDNA in ALS pathophysiology.

## 1. Introduction

Amyotrophic lateral sclerosis (ALS) is a neurodegenerative disease that irreversibly causes death of the lower motor neurons in the brainstem and spinal cord and of the upper motor neurons in the motor cortex. Mitochondrial dysfunction represents one of the critical factors in the pathogenesis of ALS [1,2]. Unlike nuclear DNA, mitochondrial DNA (mtDNA) lacks protective histones and robust repair systems, making it especially prone to damage from oxidative stress [3]. Such instability is particularly harmful to energy-demanding, post-mitotic cells like neurons and myocytes, which are sensitive to disruptions in the respiratory chain and ROS-induced damage [4]. As a consequence of this vulnerability, fragments of mtDNA may be released into the bloodstream, a phenomenon referred to as circulating cell-free mitochondrial DNA (Cf-mtDNA) [5], via apoptosis, necrosis, or active secretion, and may activate the immune system, acting as damage-associated molecular patterns (DAMPs) [6,7]. Resembling bacterial DNA, these fragments can trigger innate immune responses and inflammation across various diseases [5,8,9,10,11]. Interestingly, studies in ALS mouse models have demonstrated that the accumulation of misfolded proteins, such as SOD1 and TDP-43, can interfere with mitochondrial function, leading to the release of mtDNA into the cytoplasm and initiating immune responses [12,13]. Recently, Cf-mtDNA has emerged as a promising non-invasive diagnostic indicator of ongoing cellular damage and mortality, providing insight into the pathological mechanisms of neurodegenerative disorders, including Parkinson’s disease (PD) and Alzheimer’s disease (AD) [14]. Despite these findings, similar investigations in ALS patients are scarce [15]. Although neurofilaments are well-established and sensitive biomarkers for ALS [16,17,18], their limited specificity across neurodegenerative conditions [19] and their inability to reflect the full biological complexity of the disease highlight the need for additional, complementary biomarkers [20,21]. In this context, we investigated Cf-mtDNA levels in the serum of ALS patients compared to healthy controls and examined their association with other biomarkers and clinical measures of disease progression.

## 2. Materials and Methods

### 2.1. Study Population

ALS patients were recruited from among residents of the Emilia-Romagna region, Northern Italy, who presented to the ALS Center of the Modena University Neurological Department between 1 January 2018 and 31 December 2023. Eligible participants were considered all patients who received a diagnosis of definite or probable ALS according to El Escorial-revised criteria [22] and who still had at least 0.5 mL of serum available for analysis. ALS diagnosis was confirmed by subsequent application of the Gold Coast Criteria [23], after reviewing the medical histories of each individual patient. Serum samples were collected and biobanked as part of routine diagnostic and research procedures. Healthy control (HC) participants were recruited from the same clinical center. They consisted of either first-degree relatives of ALS patients or healthy volunteers, including blood donors, and were age- and sex-matched to the ALS group to ensure demographic comparability and reduce potential environmental bias.

### 2.2. Standard Protocol Approvals, Registrations, and Patient Consents

This retrospective exploratory study was approved by the Ethical Committee of Area Vasta Emilia Nord (file number 309/2024/TESS/AOUMO). All participants signed written informed consent for venipuncture and gave permission for the use of their biological specimens, which were stocked in the Neurobiobank of Modena (NBBM), for future research studies.

### 2.3. Clinical Measures

We collected the following demographic and anthropometric variables for each patient: sex, age at symptom onset, and weight loss at sampling. The onset of disease was determined based on the patient-reported occurrence of a distinct motor impairment in a single district [24]. Subsequently, we considered the site of onset (bulbar, spinal) and clinical phenotype (flail, upper motor neuron predominant (UMNp), bulbar, respiratory, classic) [25]. All patients started riluzole treatment after ALS diagnosis. Genetic analysis was performed for 49 patients, following previously described protocols [26]. Clinical variables used as proxies for disease progression included respiratory function, as assessed by FVC at sampling; King’s ALS staging and ALS Milano–Torino Staging (ALS-MiToS) at sampling and at last observation; disease progression rate (DPR), calculated as the monthly decline in the ALSFRS-R score from onset to diagnosis, assuming a total score of 48 at onset [27], as well as at the time of the sampling and at the last observation [28]. Slow, intermediate, and fast progressors were categorized according to DPR at sampling below 0.50, between 0.5 and 1, and above 1, respectively. We also considered time to non-invasive ventilation (NIV), invasive ventilation (IV), percutaneous endoscopic gastrostomy (PEG). Tracheostomy-free survival was defined as the timespan between disease onset and date of death or tracheostomy, whichever came first.

### 2.4. Sample Collection and Laboratory Assays

Serum samples were obtained via venipuncture and processed according to standard procedures. Following centrifugation for 10 min at 1300× *g*, the supernatant was aliquoted in polypropylene tubes and stored at −80 °C until analysis. Quantification of Neurofilament Light Chain (NfL), Neurofilament Heavy Chain (NfH), chitinase-3-like-1 (CHI3L1), SerpinA1, and Triggering Receptor Expressed on Myeloid cells 2 (TREM2) was performed using an automated next generation ELISA, based on Ella Simple Plex assay technology (BioTechne, ProteinSimple, San Jose, CA 95134, USA) following the manufacturers’ instructions [18,29]. In this immunoassay samples run through a channel each composed of three glass nano reactors pre-coated with a capture antibody, allowing for automated triplicate measurements of each sample. Samples were loaded into the cartridges with the following dilutions: 1:200,000 for the serum SerpinA1 cartridge; 1:2 for the NfL cartridge; 1:2 for the serum pNfH cartridge; 1:10 for the serum CHI3L1 cartridge; 1:10 for the serum TREM2. Intra-assay and inter-assay variability were evaluated by the manufacturer. Blood samples were analyzed at the Central Laboratory of the Azienda Ospedaliero-Universitaria di Modena using a DxH 900 analyzer (Beckman Coulter, Brea, CA, USA), a fully automated hematology system for white blood cell (WBC) counts and five-part leucocyte differential counting, and red blood cell (RBC) counts. The analyzer employs volume, conductivity, and scatter parameters for controlled flow cytometric analysis of WBC differential. The analytical software is self-gating and separates WBC populations by automatic logic pathways. All analyses were performed in accordance with the manufacturer’s instructions. The neutrophil-to-lymphocyte ratio (NLR), monocyte-to-lymphocyte ratio (MLR), systemic inflammation response index (SIRI), aggregate systemic inflammation index (AISI) and systemic–immune–inflammation index (SII) were derived from the complete blood count.

### 2.5. DNA Extraction and mtDNA Quantification Through Digital Droplet PCR

Total DNA was extracted from serum using the DNeasy Blood & Tissue Kit (QIAgen, Hilden, Germany) according to the manufacturer’s instructions, and the DNA quantity was assessed using the NanoDrop ND-1000 (Thermo Fisher Scientific, Waltham, MA, USA). mtDNA quantification was performed using droplet digital PCR (ddPCR). Each ddPCR reaction mixture (final volume of 20 μL) consisted of 10 ng of DNA samples, 2X ddPCR Supermix for Probes (Bio-Rad), and 1.1 μL of each mtDNA custom assay (Bio-Rad Laboratories, Hercules, CA, USA). The thermal protocol conditions were set as follows: initial denaturation at 95 °C for 10 min, followed by 40 cycles of denaturation at 94 °C for 30 s and annealing/extension at 55 °C for 1 min, and a final extension at 98 °C for 10 min. The droplets were then read using the QX200 ddPCR droplet reader (Bio-Rad), and analysis was performed using QuantaSoft Analysis software (version 1.7.4.0917, Bio-Rad). The wet-validated primers used were as follows: ND2 (Bio-Rad, #10031252) and Actb (Bio-Rad, #10042961).

### 2.6. Statistical Methods

Continuous variables were reported as means ± standard deviations (SDs), medians, and interquartile ranges (IQRs), whereas categorical variables were reported as absolute numbers and percentages. The use of non-parametric methods, when possible, was preferred due to the low sample sizes and because visual inspection revealed non-adherence of most of the biomarkers, including Cf-mtDNA, to the Gaussian distribution. We compared Cf-mtDNA between ALS patients and the HC group using the two-tailed Wilcoxon test. A *p*-value < 0.05 was considered statistically significant. Correlations between biomarker numerical variables were assessed using the Spearman rank correlation coefficient. Receiver operating characteristic (ROC) analysis was used to evaluate the overall performance to discriminate between ALS patients versus HC, by measuring the area under the curve (AUC). The relationship between Cf-mtDNA and clinical variables was assessed with linear (for continuous variables) or Poisson (for discrete variables) regression models, and the results were expressed as the mean difference (MD) or mean ratio (MR) related to a Cf-mtDNA increase of a half million copies per mL of serum. The relationship between Cf-mtDNA and time-to-event variables was assessed with the unadjusted Cox proportional hazards regression model. Cox analysis results were expressed as the hazard ratio (HR) related to a Cf-mtDNA increase of a half million copies per mL of serum. Uncertainty in association measures was reported with the 95% confidence interval (CI). Data analysis was performed using STATA 17 (StataCorp.2017.College Station, TX, USA) and R 3.4.3 (The R Foundation for Statistical Computing, Wien, Austria) statistical software.

## 3. Results

Based on the inclusion criteria, the serum from 54 ALS patients (27 males, 27 females, mean age at sampling: 52.88 ± 13.28 years) and 36 HC (19 male, 17 females, mean age at sampling: 51.23 ± 10.49 years) were available for analysis. ALS patients’ features are summarized in Table 1. The average value of serum NfL was 116.88 ± 68.22 pg/mL, and the average creatinine was 0.71 ± 0.20 mg/mL.

### 3.1. Biomarkers’ Distribution in ALS Patients and Healthy Controls

The average serum Cf-mtDNA was not statistically different between ALS patients and HC (copies per mL of serum: 2,426,315 ± 2,793,092, IQR: 865,000–2,475,000 vs. 1,885,667 ± 2,194,552, IQR: 394,250–2,492,500, *p* = 0.308) (Figure 1). When analyzing Cf-mtDNA discriminant ability, ROC analysis showed an AUC of 0.595 (95% CI: 0.468–0.721). Mean values for each biomarker in the ALS cohort are visible in Appendix A.

### 3.2. Cf-mtDNA Association with Clinical Indicators of Progression in ALS

We evaluated potential associations between Cf-mtDNA serum levels and ALS clinical features, as detailed in Table 2. No significant associations were found between Cf-mtDNA concentrations and any of the clinical variables analyzed. No statistically significant differences were observed in relation to FVC at the time of sampling or to clinical staging scores, either at sampling or at the last follow-up. Similar non-significant trends were also noted for ALSFRS-R scores and disease progression rates at both time points. When stratifying by genotype, comparisons between C9orf72 expansion carriers and either non-mutated (WT) or other mutation groups (non-C9orf72) (Figure 2A), as well as multi-group analyses across different genetic subtypes, revealed no significant differences in Cf-mtDNA serum levels. We also stratified the data based on disease progression rate (slow, intermediate, or fast) or site of onset (bulbar or spinal) without finding significant differences between these groups and HC (Figure 2B). Finally, no significant associations were identified between Cf-mtDNA levels and comorbidities or demographic variables among ALS patients, as reported in Appendix A.

### 3.3. Correlation of Cf-mtDNA with ALS Biomarkers

A complete list of biomarker distributions among ALS patients is reported in Appendix A. In the present analysis, Cf-mtDNA levels did not show significant correlations with well-established ALS biomarkers, as detailed in Table 3. Notably, Cf-mtDNA showed an inverse correlation with serum creatinine levels (r = –0.280, *p* = 0.0442). Finally, no significant correlations were observed between Cf-mtDNA and inflammatory indices, such as NLR, MLR, SIRI, AISI, and SII and lipid metabolism-related markers.

### 3.4. Cf-mtDNA and Survival

Cox analysis was used to assess the prognostic role of each clinical variable and biomarker in our cohort (Table 4). In the survival analysis, Cf-mtDNA levels demonstrated no significant association with key clinical outcomes in patients with ALS, including death, tracheostomy-free survival, and the use of supportive interventions, such as NIV, PEG, and IV.

## 4. Discussion

In this exploratory study, we explored serum Cf-mtDNA levels in ALS patients, assessing their diagnostic and prognostic potential in comparison with age- and sex-matched healthy controls. Cf-mtDNA concentrations did not differ significantly between ALS patients and the controls and showed no diagnostic discriminative power, which is in contrast to findings from a previous study [30]. Several potential factors may account for the discordance between these studies. First, while both studies analyzed Cf-mtDNA in serum, differences in sample preparation protocols, Cf-mtDNA quantification techniques, and cohort composition may have contributed to the observed discrepancies. It is important to note that Cf-mtDNA quantification can be highly sensitive to pre-analytical variables, particularly the DNA extraction method. In our study, we used a standardized protocol across all samples, employing the kit and protocol, which has been previously reported to offer the most consistent performance in Cf-mtDNA recovery [31]. In particular, Li et al. used a different quantitative assay, which may have affected their sensitivity and specificity in detecting Cf-mtDNA variations. Moreover, differences in sample processing and storage duration may introduce pre-analytical variability, which is increasingly recognized as a major challenge in the field of CfDNA-based biomarker discovery. Also, in Li’s study, a genetically distinct cohort enriched with SOD1 mutation carriers was included, in whom they observed the highest Cf-mtDNA levels and the strongest associations with disease progression. ALS exhibits high clinical and genetic heterogeneity; thus, subgroup-specific patterns may be masked in aggregated analyses, particularly in smaller and mainly sporadic cohorts like ours. Interestingly, while Li et al. identified an inverse correlation between Cf-mtDNA and ALS progression rate, particularly in SOD1 mutation carriers, we found no association between Cf-mtDNA and clinical measures of disease severity or survival. SOD1-associated ALS may be characterized by pronounced mitochondrial involvement [32], potentially leading to increased systemic Cf-mtDNA release.

Moreover, Cf-mtDNA was not significantly associated with core biomarkers of neurodegeneration and inflammation. Interestingly, a significant inverse correlation was identified between Cf-mtDNA levels and serum creatinine, a finding that warrants further investigation. While creatinine is commonly used as a proxy for muscle mass and renal function, its association with Cf-mtDNA could suggest shared underlying mechanisms related to mitochondrial metabolism or systemic catabolism. However, this relationship must be interpreted with caution given the complexity of metabolic alterations in ALS and the potential influence of comorbidities. In this context, elevated Cf-mtDNA has been linked to oxidative stress and systemic inflammation, while reduced creatinine levels—when reflecting malnutrition or sarcopenia—may serve as indicators of advanced disease stage. Both markers, therefore, could represent distinct yet converging facets of the same pathological trajectory. Furthermore, the reduction in serum creatinine commonly observed in ALS is consistent with progressive muscle atrophy. At the same time, ALS may cause progressive muscle atrophy, in this way inducing ongoing muscle damage and increased mitochondrial turnover that promotes the release of Cf-mtDNA into the circulation, potentially contributing to the inverse association observed in our cohort. Next, it is worth noting that a previous study reported a correlation between urinary mtDNA levels and creatinine in patients with hypertension-related kidney dysfunction [33]. While this evidence pertains to a different biological matrix and clinical context, it may support the plausibility of a link between mitochondrial nucleic acids and creatinine metabolism.

In addition to ALS, Cf-mtDNA alterations have been reported in other neurodegenerative disorders. In the case of multiple sclerosis (MS) [34], elevated levels were associated with disease activity and neuroinflammation, while no differences were observed in cases of PD and multiple system atrophy [35]. The picture is more complex as far as AD is concerned, as mtDNA levels have been shown to be lower than controls, but deeply influenced by environmental factors [36,37]. A recent review by Aydın et al. [38], emphasized the diagnostic utility of Cf-mtDNA in conditions such as AD and PD, where elevated Cf-mtDNA levels, when observed, could be interpreted as a surrogate marker of ongoing neuroinflammation and mitochondrial dysfunction. In ALS, however, the absence of a significant increase in Cf-mtDNA, along with the lack of correlation with established biomarkers of neurodegeneration and inflammation, suggests a disease-specific behavior of this biomarker. One possible explanation lies in the distinct pathophysiological mechanisms of ALS. Although mitochondrial dysfunction is well recognized in ALS, it may lead to a more compartmentalized or cell-type-specific release of Cf-mtDNA, limiting its detectability in peripheral circulation. Supporting this hypothesis, Cf-mtDNA levels in our cohort did not correlate with neurofilaments, which are considered robust markers of neuronal damage in ALS and other neurodegenerative conditions. This lack of association may imply that Cf-mtDNA reflects a separate pathophysiological axis, potentially more related to metabolic or systemic stress than to direct neuroaxonal injury. Another key consideration is the biological matrix. While Aydın et al. discuss Cf-mtDNA derived from both plasma and CSF [38], our study is limited to peripheral blood serum. It is conceivable that Cf-mtDNA dynamics in ALS are more prominent within the central nervous system or in neuron-derived vesicles, and are thus underrepresented in peripheral blood.

Future studies using CSF samples or cell-specific profiling approaches—such as analysis of neuronal exosomes—will be essential to clarify the source and relevance of Cf-mtDNA in ALS pathophysiology.

The present study has some strengths and limitations. All ALS patients were deeply clinically characterized and followed up, and a broad panel of inflammatory and neurodegeneration biomarkers allowed for a multifaceted examination of Cf-mtDNA relevance within ALS pathophysiology. Survival data and a wide list of clinical indicators constitute a significant added value to our work. Moreover, healthy controls were age- and sex-matched with our ALS population, therefore reducing the risk of confounding bias. However, the sample size, although comparable to similar exploratory studies, may have been underpowered to detect small-to-moderate associations. Another critical limitation affecting the clinical translatability of our findings is the absence of a replication cohort and cerebrospinal fluid (CSF) data, which restricts the assessment of compartment-specific dynamics and limits generalizability. Finally, the design of the study precludes conclusions about the longitudinal dynamics of Cf-mtDNA throughout disease progression. Despite these limitations, our study offers insights by showing that Cf-mtDNA, at least in serum, may not reflect disease severity, inflammation, or survival in ALS. The observed inverse correlation with creatinine is intriguing and may point toward a link with muscle mass and systemic catabolism, both of which are altered in ALS progression. However, this finding requires further exploration.

## 5. Conclusions

In conclusion, our findings suggest that Cf-mtDNA levels in serum are not significantly altered in ALS nor associated with disease progression or survival. We could not find correlations with other biomarkers of neurodegeneration or neuroinflammation. Although a potential link with serum creatinine levels was observed, the overall data suggest that Cf-mtDNA, as measured in peripheral blood, may not capture the specific neurodegenerative processes characteristic of ALS. These results contrast with previous findings in ALS and other neurodegenerative disorders, highlighting the need for further investigation into the biological matrices, temporal dynamics, and disease-specific factors that modulate Cf-mtDNA levels. Future studies should consider larger cohorts, longitudinal sampling, and the exploration of Cf-mtDNA in different biofluids to better understand its compartment-specific relevance and mechanistic roles in ALS. Our work emphasizes the importance of harmonizing analytical methods and considering disease compartmentalization in future studies.

## Figures and Tables

**Figure 1 cells-14-01433-f001:**
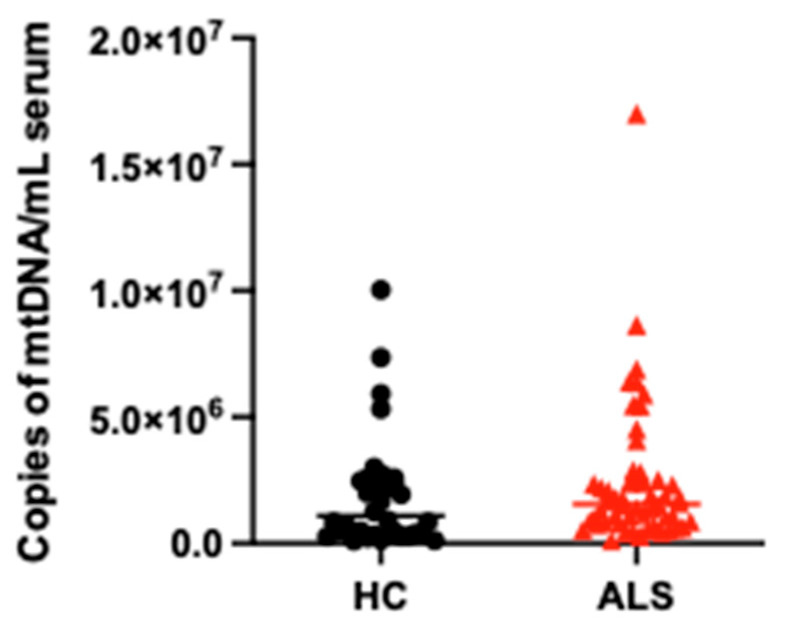
Cf-mtDNA distribution in serum from ALS patients (red) and HC (black). Scatter dot plots of Cf-mtDNA concentrations in serum of ALS patients and HC.

**Figure 2 cells-14-01433-f002:**
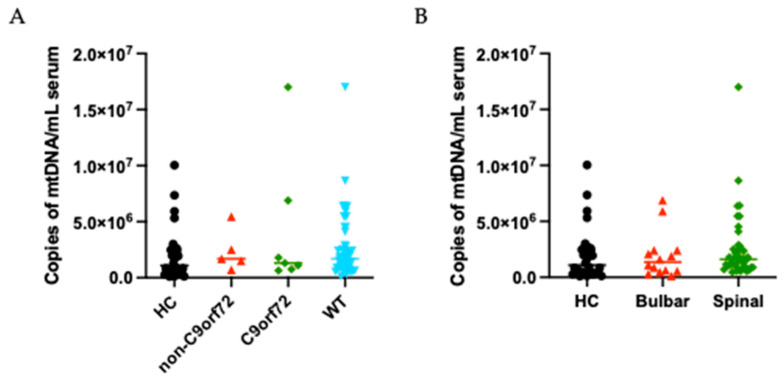
Cf-mtDNA distribution in serum from ALS patients stratified by genotype and site of onset. (**A**) Scatter dot plots of Cf-mtDNA concentrations in serum of C9orf72 mutated ALS patients (green), non-C9orf72 ALS patients (red), WT ALS patients (light blue), and HC (black). (**B**) Scatter dot plots of Cf-mtDNA concentrations in serum of bulbar ALS patients (red), spinal ALS patients (green), and HC (black).

**Table 1 cells-14-01433-t001:** Demographic and clinical characteristics of ALS patients in the study.

Variable	Patients (n = 54), n (%), Mean [SD] Median [IQR]
Age at onset, years	52.03 [13.15] 50.47 [40.74–62.70]
Weight loss at sampling, Kg	2.70 [6.09] 0.50 [0.00–3.75]
Mutational status †	
*C9ORF72*/other/WT	7 (14.3%)/5 (10.2%)/40 (74.1%)
Site of onset	
Bulbar, spinal	14 (25.9%)/40 (74.1%)
Phenotype	
Flail, UMNp, bulbar, classic	8 (14.8%)/3 (5.6%)/9 (16.7%)/34 (63.0%)
ALSFRS-r total score at sampling, points	40.87 [4.39] 42.00 [38.25–44.00]
DPR at sampling, points/month	1.25 [2.16] 0.65 [0.36–1.21]
MiToS score at sampling	0.15 [0.36] 0.00 [0.00–0.00]
King’s staging at sampling	1.72 [0.79] 2.00 [1.00–2.00]
FVC at sampling, %	94.24 [20.02] 91.50 [79.75–106.00]
NIV	34 (63.0%)
PEG	29 (53.7%)
IV	22 (40.7%)
ALSFRS-r total score at last observation, points	14.85 [10.58] 14.00 [6.00-20-00]
DPR at last observation, points/month	1.12 [0.90] 0.84 [0.70–1.26]
MiToS score at sampling	2.60 [1.25] 2.00 [2.00–4.00]
King’s staging at sampling	3.49 [0.72] 4.00 [3.00–4.00]
Comorbidities	
Depression/psychosis	15 (27.8%)/2 (3.7%)
COPD/other respiratory disease	4 (7.4%)/5 (9.3%)
Diabetes	4 (7.4%)
Hypertension	20 (37.0%)
Cardiopathies	6 (11.1%)
Dyslipidemia	16 (29.6%)
Autoimmune diseases	4 (7.4%)
Oncological history	5 (9.3%)

**Notes**: Means with standard deviations [SDs], medians, and interquartile ranges (IQRs) are reported for numerical variables and absolute numbers with percentages (%) for categorical variables. † Genetic analysis available for 49 patients. **Legend**: COPD: Chronic Obstructive Pulmonary Disease. SD: standard deviation. UMNp: Upper Motor Neuron predominant. WT: wild-type. ALSFRS-r: Amyotrophic Lateral Sclerosis Functional Rating Scale—revised. DPR: disease progression. FVC: forced vital capacity. NIV: non-invasive ventilation. PEG: percutaneous endoscopic gastrotomy. IV: invasive ventilation.

**Table 2 cells-14-01433-t002:** Associations between Cf-mtDNA and clinical variables of disease progression in ALS patients in the study.

Variable	Association Measure (95% CI)	*p*-Value
FVC at sampling	MD = 0.074 (−0.900, 1.048)	0.8822
ALSFRS-R at sampling	MD = −0.040 (−0.253, 0.174)	0.7166
DPR at sampling	MD = −0.045 (−0.150, 0.059)	0.3993
MiTos at sampling	MR = 0.989 (0.862, 1.135)	0.8750
King’s staging at sampling	MR = 1.007 (0.973, 1.042)	0.7020
ALSFRS-R at last observation	MD = −0.200 (−0.718, 0.318)	0.4529
DPR at last observation	MD = −0.002 (−0.047, 0.042)	0.9144
MiTos at last observation	MR = 1.010 (0.982, 1.038)	0.5006
King’s staging at last observation	MR = 1.004 (0.979, 1.029)	0.7830

**Legend**: ALSFRS-r: Amyotrophic Lateral Sclerosis Functional Rating Scale—revised. DPR: disease progression. FVC: forced vital capacity. MD = mean difference. MR = mean ratio. CI = confidence interval. Association measures are related to a Cf-mtDNA increase of a half million copies per mL of serum.

**Table 3 cells-14-01433-t003:** Correlations between Cf-mtDNA concentrations and other biomarkers.

Variable	Spearman Correlation	*p*-Value
NfL_serum_	0.022	0.8774
pNfH_serum_	−0.054	0.7022
SerpinA1_serum_	−0.110	0.4438
TREM2_serum_	0.185	0.2078
CHIT3L1_serum_	−0.053	0.7109
NLR	−0.043	0.7625
MLR	0.074	0.5967
SIRI	0.038	0.7879
AISI	0.043	0.7603
SII	0.001	0.9961
Total Cholesterol	−0.046	0.7493
HDL Cholesterol	−0.166	0.2592
LDL Cholesterol	−0.080	0.5884
Triglycerides	−0.015	0.9164
Creatinine	−0.280	0.0442

**Legend:** NLR: neutrophil-to-lymphocyte ratio, MLR: monocyte-to-lymphocyte ratio, SIRI: systemic inflammation response index, AISI: aggregate systemic inflammation index, SII: systemic–immune–inflammation index.

**Table 4 cells-14-01433-t004:** Association of Cf-mtDNA concentrations with survival outcomes.

Outcome	HR (95% CI)	*p*-Value
NIV	1.005 (0.935–1.079)	0.8991
PEG	1.020 (0.970–1.074)	0.4332
IV	1.021 (0.967–1.078)	0.4485
Death	1.016 (0.967–1.067)	0.5279
Tracheostomy-free survival	1.016 (0.974–1.059)	0.4665

**Notes:** Hazard ratios are presented with 95% confidence interval and p-value. CI: confidence interval. HR: hazard ratio. NIV: non-invasive ventilation. PEG: percutaneous endoscopic gastrotomy. IV: invasive ventilation. Hazard ratios are related to a Cf-mtDNA increase of a half million copies per mL of serum.

## Data Availability

Data are available from the authors upon reasonable request and after providing the approval of the ethical committee.

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
