# Peer review of "Circulating Serum Cell-Free Mitochondrial DNA in Amyotrophic Lateral Sclerosis"

_cells, 2025, doi:10.3390/cells14181433_

Round 1

Reviewer 1 Report

Comments and Suggestions for Authors

The paper aims to reveal the correlation between the cell-free mitochondrial DNA amount in serum and amyotrophic lateral sclerosis (ALS). The study is well designed, data are treated on a good level and results are reliable. The main conclusion of the study is the absence of association between cell-free mitochondrial DNA amount and amyotrophic lateral sclerosis. Interestingly, in addition no significant associations of cell-free mitochondrial DNA was revealed  with ALS phenotype, disease staging, neurofilaments, inflammatory indices, or survival. Only one significant correlation shows inverse relationship of cell-free mitochondrial DNA with serum creatinine levels.

The main result of study (no correlation of ALS with the cell-free mitochondrial DNA) contradicts results from other study (Li et al., 2024) and in the well written “Discussion” authors are trying to support their conclusion and explain the possible reasons of contradiction. The Li et al (2024) have used PCR with ND-1 primers, whereas the present study with ND-2 primers. These two genes are adjacent in mitochondrial DNA and I do not think that it could be the reason of controversy. As authors are writing cited paper included genetically distinct cohort enriched with SOD1 mutation carriers, which could be one of the reasons for discrepancy.

In conclusion, it is a well conducted, analyzed and written important paper, which deserves to be published

Author Response

Response to Reviewer 1 Comments

We thank the Reviewer for the thoughtful and thorough evaluation of our revised manuscript. We greatly appreciate the constructive feedback, which has helped us to further improve the quality and clarity of our work.

Please find below our point-by-point responses to each of the reviewer' comments

Ilaria Martinelli and Elisabetta Zucchi

2. Questions for General Evaluation

Reviewer’s Evaluation

Response and Revisions

Does the introduction provide sufficient background and include all relevant references?

Yes

We sincerely thank the reviewer for the positive evaluation of our paper and for the valuable suggestion to further enhance the manuscript.

Are all the cited references relevant to the research?

Yes

Is the research design appropriate?

Yes

Are the methods adequately described?

Yes

Are the results clearly presented?

Yes

Are the conclusions supported by the results?

Yes

3. Point-by-point response to Comments and Suggestions for Authors

The paper aims to reveal the correlation between the cell-free mitochondrial DNA amount in serum and amyotrophic lateral sclerosis (ALS). The study is well designed, data are treated on a good level and results are reliable. The main conclusion of the study is the absence of association between cell-free mitochondrial DNA amount and amyotrophic lateral sclerosis. Interestingly, in addition no significant associations of cell-free mitochondrial DNA was revealed  with ALS phenotype, disease staging, neurofilaments, inflammatory indices, or survival. Only one significant correlation shows inverse relationship of cell-free mitochondrial DNA with serum creatinine levels.

The main result of study (no correlation of ALS with the cell-free mitochondrial DNA) contradicts results from other study (Li et al., 2024) and in the well written “Discussion” authors are trying to support their conclusion and explain the possible reasons of contradiction. The Li et al (2024) have used PCR with ND-1 primers, whereas the present study with ND-2 primers. These two genes are adjacent in mitochondrial DNA and I do not think that it could be the reason of controversy. As authors are writing cited paper included genetically distinct cohort enriched with SOD1 mutation carriers, which could be one of the reasons for discrepancy.

In conclusion, it is a well conducted, analyzed and written important paper, which deserves to be published

Response:

We thank the Reviewer for this important point. We agree with her/his considerations: the discrepancy with Li et al. (2024) is unlikely to be due to the use of ND1 vs ND2 primers, since these genes are adjacent in the mitochondrial genome. Instead, it likely reflects methodological differences (qPCR with standard curves vs ddPCR, DNA extraction procedures, reporting units) and, above all, the distinct cohort composition: our study mainly included sporadic ALS, whereas Li et al. investigated an SOD1-enriched cohort, in which cf-mtDNA levels were particularly elevated. We have clarified this point in the revised Discussion to improve transparency and facilitate comparison between studies (lines 271-289).

Reviewer 2 Report

Comments and Suggestions for Authors

Zanini and colleagues provided a study on circulating cell-free mitochondria DNA in serum of amyotrophic lateral sclerosis patients.  Using droplet digital PCR (ddPCR), they revealed no significant differences between healthy controls and ALS patients. At the current state, there are several major concerns with the manuscript.

1) Tile need to include ‘serum’ so readers know which biofluids was used and should somehow also mention the ALS biomarkers.

2) Unclear why it is pilot study? I think it is not necessary albeit most of your data is not significant.

3) Line 54-55. What evidence suggest cf-mtDNA are from neurons and glia and not other cell types within the brain (e.g. epithelial cells) or peripherally since even if used CSF as mentioned in the article, the BBB is compromise in ALS.

4) Method section. Unclear why serum was initially collected as it seems it was not for this mitochondria study. Need to state if the healthy controls were staff of the clinical centre or non-ALS patients or family member of the ALS patients? The latter may be the most appropriate group as the exposure to potential environmental factors would be  similar to the ALS patient than other groups.

5) More information required in statistical section. How was data determined if data was normally distributed or not? Similarly, why Spearman’s and not Pearson’s correlation was used.

6) I’m sure the authors are aware of the heterogeneity of the disease and the study has large SD in the DPR at sampling. If so, they need to reanalyse the data. For example, there are individuals with fast or slow onset, so need to provide a separate graph for this analysis.  Also, need to another graph with separate ‘definite’ and ‘probable’ groups in analysis.

7) Unclear how figure 1 is different to table 2, since it’s same data? If so, just keep the figure.

8) Important to understand the cohort of patients used in this study, since apart from the clinical data, there are no ALS molecular data to support this study. Therefore, need to show the ALS biomarker data (e.g. NfL and creatinine) graphs alone without the cf-mtDNA. Only then will we see if the cohort of patients used in this study is a typical or a unique ALS group.

9) Need to reanalyse the ALS data but separating it into 3 groups: C9orf72, other, WT, since authors question another study using on SOD1 mutation carriers only. Alternatively, can use a point on graph for each single patient, then colour code these points based on the subgroups.

10) In the discussion, there are a lot of comparison with a single study to try and explain why this study did not detect significant differences between the groups. However, in the discussion it was unclear: 1) how sample preparation would affect the cf-mtDNA? And 2) how was the previous study making the detection of cf-mtDNA more sensitive since ddPCR is a highly sensitive technique.

11) Should comments on other things in discussion that focus on comparison with other published papers and just one paper. 

Minor concerns:

1) The values in thousands doesn’t need to have 1 decimal point.

2) Line 49-50. Is mention DAMPs more appropriate?

3) Line 110. Unclear the meaning of ‘for the first each control was tested 16 times’.

4) Table 1 legend. Missing definition for ‘WT’

5) Line 227. It seems to be only one study not ‘studies’

Reviewer 3 Report

Comments and Suggestions for Authors

This manuscript compared the copy number of serum cf-mtDNA between ALS patients and healthy people, and showed no significant difference. The results suggested that the level of cf-mtDNA in serum couldn’t be used as diagnostic biomarker of ALS. This conclusion is contrary to previous reports. In the previous report (PMID: 39324867), the number of cf-mtDNA copies is about ten times more than the number in this manuscript. The authors should explain the difference.
